# Reference values for respiratory muscle strength and maximal voluntary ventilation in the Brazilian adult population: A multicentric study

**Palomma Russelly Saldanha Araújo**[1,2], **Jéssica Danielle Medeiros da Fonseca**[1,2], **Ana Aline Marcelino**[1,2], **Marlene Aparecida Moreno**[3], **Arméle de Fátima Dornelas de Andrade**[4], **Matias Otto Yañez**[5], **Rodrigo Torres-Castro**[6], **Vanessa Regiane Resqueti**[1,2], **Guilherme Augusto de Freitas Fregonezi**[1,2]*

1 PneumoCardioVascular Lab/HUOL, Departamento de Fisioterapia, Hospital Universitário Onofre Lopes, Universidade Federal do Rio Grande do Norte, Natal, Rio Grande do Norte, Brazil, 2 Laboratório de Inovação Tecnológica em Reabilitação, Departamento de Fisioterapia, Universidade Federal do Rio Grande do Norte Natal, Rio Grande do Norte, Brazil, 3 Laboratório de Pesquisa em Performance Humana, Universidade Metodista, Piracicaba, São Paulo, Brazil, 4 Laboratório Fisioterapia CardioPulmonar, Departamento de Fisioterapia, Universidade Federal de Pernambuco, Recife, Pernambuco, Brazil, 5 Esculea de Kinesiologia, Universidad Autonoma de Chile, Chile, 6 Department of Physical Therapy, University of Chile, Santiago, Chile

* guilherme.fregonezi@ufrn.br

**Data Availability Statement:** All relevant data are within the paper and its Supporting information.

## Abstract

### Aim

To determine reference values and propose prediction equations for respiratory muscle strength, maximal inspiratory pressure (MIP), maximal expiratory pressure (MEP), and endurance by means of maximal voluntary ventilation (MVV) in healthy Brazilian adults.

### Methods

Anthropometric data, level of physical activity, pulmonary function, and respiratory muscle strength and maximal voluntary ventilation of 243 participants (111 men and 132 women) aged 20 to 80 years were assessed at three cities in the southeast and northeast region of Brazil.

### Results

Mean maximal respiratory pressures and MVV were higher in men (MIP = 111.0 ± 28.0; MEP = 149.6 ± 40.3; MVV = 150.6 ± 35.2) than in women (MIP = 87.9 ± 17.6; MEP = 106.7 ± 25.2; MVV = 103.4 ± 23.2; all p < 0.05). Based on regression models, the following prediction equations were proposed for men: MIP = 137–0.57 (age), $R^2$ = 0.13, standard error of estimate (SEE) = 26.11; MEP = 179.9–0.67 x (age), $R^2$ = 0.08, SEE = 38.54; and MVV = 206.3–1.18 x (age), $R^2$ = 0.36, SEE = 28.08. Prediction equations were also proposed for women: MIP = 107.3–0.4 x (age), $R^2$ = 0.16, SEE = 16.10; MEP = 127.4–0.43 x (age), $R^2$ = 0.08, SEE = 24.09; and MVV = 146.3–0.86 x (age), $R^2$ = 0.42, SEE = 17.56.

**Funding:** The author(s) received specific funding for this work: Coordenação de Aperfeiçoamento de Pessoal de Nível Superior (CAPES), Finance Code 001 - Ms. Ana Aline Marcelino Conselho Nacional de Desenvolvimento Científico e Tecnológico (CNPq), 316937/2021-5 - Mr. Guilherme A. F. Fregonezi Conselho Nacional de Desenvolvimento Científico e Tecnológico (CNPq), 305960/2021-0 - Ms. Vanessa Regiane Resqueti Coordenação de Aperfeiçoamento de Pessoal de Nível Superior (CAPES), 88887.692599/2022-00 - Ms. Jéssica Danielle Medeiros da Fonseca.

**Competing interests:** The authors have declared that no competing interests exist.

## Conclusion

Reference values for MIP, MEP, and MVV were determined in healthy Brazilian adults. Results from different Brazilian regions provided adequate prediction equations considering an ethnically heterogeneous population.

## Introduction

The assessment of pulmonary function, strength and maximal voluntary ventilation of respiratory muscles allows monitoring respiratory diseases, early detection of respiratory muscle weakness and fatigue, and prognostic and predictive information on patient survival [1–4].

Respiratory muscle strength is commonly assessed using maximal respiratory pressure (MRP) (i.e., maximal inspiratory pressure [MIP] and maximal expiratory pressure [MEP]) [5]. Respiratory muscle endurance is assessed using maximal voluntary ventilation (MVV) [6]. The validity of MVV in assessing respiratory endurance during brief 12 or 15-second test is uncertain [5, 6]. Nevertheless, this maneuver assesses maximum ventilatory capacity, reflecting the functioning of the inspiratory pump and chest wall [7]. It is actually clinically utilized the most to determine ventilatory reserve [8, 9], assess risk of postoperative complications [10] and establish targets for muscle training [11]. These effort-dependent and noninvasive assessments are easy to apply and commonly used to diagnose and monitor patient progression [12, 13].

Reference values allow comparing assessment results and expected values according to specific biological characteristics of a population [14]. Nevertheless, technical, individual, and population differences may lead to variability in reference values and prediction equations [13]. Although previous studies suggested prediction equations for MIP, MEP, and MVV in the Brazilian population [12, 15–17], were based on the local population, not multicenter studies. Moreover, there is methodological variability regarding MIP, MEP, and MVV data. Many MIP and MEP studies used less accurate and not recommended mechanical devices, since the nineties, to study respiratory muscle strength [12, 16, 17]. In fact, guidelines recommend the use of a digital manometer with high precision [18]. Regarding MVV, the only study that offers reference values for this variable was conducted at a single center [12]. Therefore, there is a need to update these values.

We aim to establish new reference values for MIP and MEP, as well as MVV, in a diverse sample of healthy Brazilian adults aged 20 to 80 years, residing in three different cities in Brazil. In addition to these primary objectives, our research endeavors to explore potential variations in these respiratory parameters across sex and age groups. By doing so, we aim to not only provide updated and relevant benchmarks for clinical and research purposes but also to propose comprehensive prediction equations specifically tailored to the unique characteristics of the Brazilian population. These new equations will improve our understanding of respiratory health and provide better guidance for diagnostic and treatment approaches. In addition, we will compare the values resulting from our generated equation with those obtained from previously published equations.

## Methods

### Study type and sample

This multicenter cross-sectional study was conducted between September 01, 2009 and December 25, 2011 at three cities in Brazil: Natal (Rio Grande do Norte), Recife

(Pernambuco), and Piracicaba (São Paulo). This study was reported following the "Strengthening the Reporting of Observational Studies in Epidemiology" (STROBE) statement [19].

Participants were recruited by convenience through publicity to university students at each center, as well as via social media and invited to participate according to the following inclusion criteria: self-reported healthy Brazilian individuals; age between 20 and 80 years; body mass index (BMI) between 18.5 and 29.9 kg/m$^2$; non-smoker; non-pregnant; without respiratory, neuromuscular, cerebrovascular, orthopedic, or cardiac diseases; with normal pulmonary function (i.e., forced vital capacity [FVC] > 80% and ratio between forced expiratory volume in the first second [$FEV_1$] and FVC > 0.7 or > 85% of predicted), without any acute respiratory symptoms in the last month [20]. Those who had difficulty performing tests or quit the study were excluded. Participants had no previous contact with respiratory muscle strength assessments.

This study was approved by the research university ethics committee numbers 260/08. All participants were informed about the objectives and methods of the study and signed written the informed consent form. The data from this study is part of an umbrella study, a portion of which was previously published [21].

## Study design

Participants were divided into six age groups (20 to 29, 30 to 39, 40 to 49, 50 to 59, 60 to 69, and 70 to 80 years), each group was divided according to sex (men and women), totaling 12 groups. Assessments were conducted in two stages. The first stages included a structured interview (sociodemographic data and previous diseases), anthropometry data (weight, height, and Body Mass Index–BMI), and spirometry. The second stage took place following a 30-minute rest period, during which participants underwent MRP and MVV assessments and completed a questionnaire on habitual physical activity (HPA). The order of respiratory assessments was randomized, respecting the 20-minute interval between them. The same previously trained evaluator at each research center conducted both stages on the same day. Fig 1 illustrates the study flowchart.

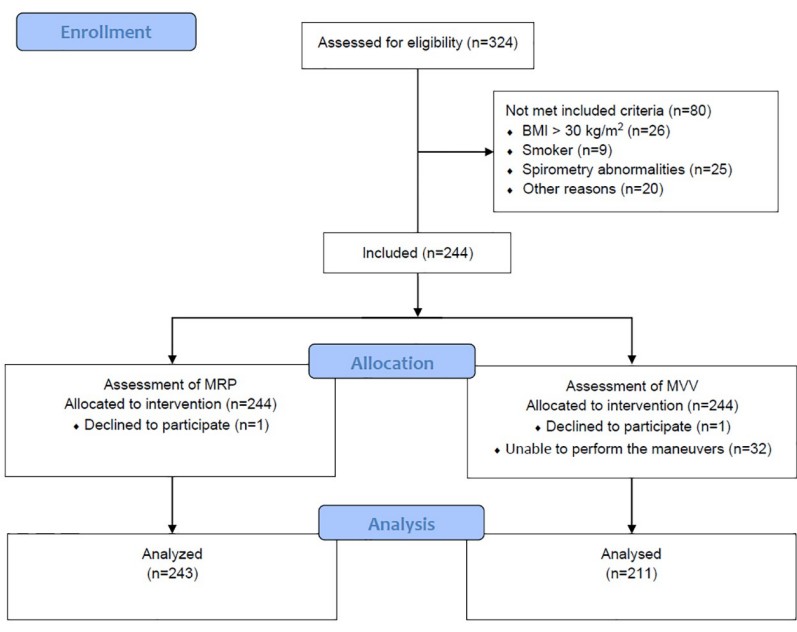

**Fig 1. Study flowchart.**

## Anthropometric variables

Weight was measured using an anthropometric scale (model 31, Filizola®, São Paulo, Brazil), whereas height was assessed during maximum inspiration using a stadiometer integrated into the scale (1 mm precision Participants were measured wearing light clothing and were barefoot. BMI was calculated considering the ratio between weight and height squared (kg/m²) and classified according to the World Health Organization (WHO) [22].

## Spirometry

Pulmonary function and endurance were assessed through FVC and MVV, respectively, using a DATOSPIR-120C® spirometer (Sibelmed®, Barcelona, Spain), according to the American Thoracic Society / European Respiratory Society (ATS/ERS) [23]. FVC, $FEV_1$, and $FEV_1$/FVC ratio were analyzed as absolute and reference values for the Brazilian population [20]. Participants performed assessments seated on a chair with backrest and feet supported on the floor.

During FVC, participants performed a maximal inspiration until total lung capacity, followed by a complete, fast, and vigorous expiration until residual volume. The assessment was repeated three to five times after one-minute intervals and considered acceptable when the difference between the two largest FVC and FEV1 values varied less than ≤0.150 L [23]. For MVV assessment, participants were asked to breathe as fast and deep as possible above the tidal volume and below the vital capacity for 15 seconds. At least two acceptable measurements were obtained (< 10% of variation), and the highest value was recorded, extrapolating the accumulated volume from 15 seconds to 1 minute [24].

## Respiratory muscle strength

Respiratory muscle strength was measured using a digital manometer (MicroRPM, CareFusion, UK), following the European Respiratory Society guidelines [5]. The PUMA PC software (Micro Medical, Rochester Kent, UK) operationalized the MIP and MEP (equivalent to MEP) variables using the aforementioned maximum mean pressures. Participants performed a maximum inspiration from residual volume for MIP and a maximum expiration from total lung capacity for MEP, performing 3–5 maneuvers, with a 1 minute break between them. Both were assessed with participants using a disposable flanged mouthpiece and a nose clip. A minimum of three reproducible and acceptable measurements were performed; if the last maneuver was the highest, a new maneuver should be performed, and variability should not exceed 10%. The highest value of maximum pressure sustained was selected.

## Level of physical activity

The level of physical activity was assessed using the Brazilian version of the Baecke Habitual Physical Activity–HPA questionnaire [25]. This version has eight questions and covers two physical activity components performed in the last 12 months: 1) physical exercises in leisure (PEL, with four questions) and 2) leisure and locomotion activities excluding physical exercises (LLA, with four questions). Answers were scored using a Likert scale (0 to 5), and PEL and LLA scores average (Q1 + Q2 + Q3 + Q4 / 4) correspond to the respective scores. The average of scores (PEL + LLA/2) represented the HPA index [25]. For analysis we select the average of values of the HPA index.

## Statistical analysis and sample size calculation

A pilot study with 29 women and 32 men was used to calculated the sample size based on the effect size ($R^2$) of regression analysis for MIP, MEP, and MVV variables found by Neder et al.

[12] and considering a power of 99% and alpha error of 0.05. These values were determined for each center (i.e., 183 participants; 87 women and 96 men) and doubled to minimize exclusions due to data acquisition errors or dropouts. The total sample size was estimated as 366 participants to be distributes across for all centers.

Data were expressed as mean ± standard deviation. The Kolmogorov-Smirnov test verified data normality. Unpaired t-test and ordinary one-way ANOVA compared MIP, MEP, and MVV between sex and age groups, respectively; the Newman-Keuls post-hoc test was used in case of statistical significance between age groups. Linear Pearson's correlation coefficient (r) assessed relationships between dependent (MIP, MEP, and MVV) and independent variables (age, weight, height, BMI, and HPA). Independent variables with statistical significance were included in the linear regressions for MIP, MEP, and MVV. Lower limits were calculated from the fifth percentile of residuals using a normal distribution according to the following equation: lower limit = predicted value– 1.645 x standard error of estimate (SEE) [18]. The SPSS software (IBM Corp., version 15.0, NY, USA) analyzed data considering $p < 0.05$ and 95% confidence intervals.

## Results

A total of 324 participants were recruited, and 244 were included in the study (Fig 1). One participant dropped out of the study; thus, 243 participants (111 men and 132 women) performed the MRP. Of these, 32 could not perform MVV (i.e non-reproducible tests, characterized by >10% variation and/or inability to perform the test in less than 12 seconds), totaling 211 participants (110 men and 111 women) for this measurement.

### Characterization of sample

Table 1 shown the mean age, anthropometric data, and HPA. Table 2 presents pulmonary function divided by sex. MIP, MEP, and MVV separated by sex and age are described in Table 3.

### Prediction equations

Age was negatively correlated with MRP and MVV in men (MIP: r = − 0.37; MEP: r = − 0.30; MVV: r = − 0.60; $p < 0.001$) and women (MIP: r = − 0.40; MEP: r = − 0.30; MVV: r = − 0.65; $p < 0.001$) (Table 4). However, the MRP of both sexes was not correlated with weight, height, BMI, and HPA. On the other hand, MVV was positively correlated with height in women (r = 0.44; p = 0.02).

Age was included in linear regressions for MIP, MEP, and MVV for men and women. The prediction equations for MRP and MVV in Brazilian adults aged between 20 and 80 years were established. The prediction equations for MRP and MVV in Brazilian adults aged between 20 and 80 years were established, and the coefficients of determination adjusted ($R^2$), standard error of estimate (SEE), and lower limit of normal (LLN) for sex are shown in Table 4.

### Comparation with previously published equations

The S1 Table describes previous studies that established prediction equations for MRP [12, 15–17] and MVV [12] in the Brazilian adult population. Also, Fig 2 shows MIP, MEP, and MVV values divided by sex and compared with reference values from previous studies.

**Table 1. Characterization of individuals who performed MRP and MVV assessments.**

| Age groups (years) | Anthropometric data | | | | |
|---|---|---|---|---|---|
| | Age (years) | Weight (Kg) | Height (m) | BMI (Kg/m$^2$) | HPA |
| | MRP | | | | |
| Men (n = 111) | | | | | |
| 20 to 29 (n = 27) | 22.1 ± 2.2 | 73.9 ± 10.6 | 1.75 ± 0.09 | 24.1 ± 2.4 | 5.2 ± 1.3 |
| 30 to 39 (n = 19) | 34.4 ± 3.4 | 76.3 ± 8.5 | 1.75 ± 0.04 | 25.0 ± 2.6 | 4.8 ± 1.3 |
| 40 to 49 (n = 19) | 44.2 ± 2.6 | 75.6 ± 9 | 1.70 ± 0.06 | 26.0 ± 2.6 | 5.0 ± 1.7 |
| 50 to 59 (n = 16) | 53.4 ± 2.8 | 77.1 ± 10.8 | 1.72 ± 0.06 | 26.0 ± 2.9 | 5.2 ± 1.2 |
| 60 to 69 (n = 16) | 63.9 ± 2.6 | 74.6 ± 9 | 1.65 ± 0.09 | 27.4 ± 1.7 | 5.1 ± 1.6 |
| 70 to 80 (n = 14) | 74.6 ± 4.0 | 71.4 ± 7.2 | 1.66 ± 0.08 | 26.0 ± 2.1 | 4.7 ± 1.0 |
| Women (n = 132) | | | | | |
| 20 to 29 (n = 25) | 23.2 ± 3.3 | 61.9 ± 10.8 | 1.65 ± 0.06 | 22.8 ± 3.3 | 4.7 ± 0.1 |
| 30 to 39 (n = 20) | 33.5 ± 3.2 | 62.6 ± 7.4 | 1.63 ± 0.05 | 23.6 ± 2.2 | 4.4 ± 1.2 |
| 40 to 49 (n = 24) | 45.1 ± 3.4 | 63.6 ± 9.7 | 1.60 ± 0.07 | 24.9 ± 3.5 | 4.5 ± 1.5 |
| 50 to 59 (n = 24) | 54.5 ± 3.5 | 63.3 ± 7.3 | 1.58 ± 0.06 | 25.5 ± 2.9 | 4.5 ± 1.3 |
| 60 to 69 (n = 19) | 64.3 ± 3.1 | 65.2 ± 8.1 | 1.57 ± 0.07 | 26.5 ± 2.8 | 4.6 ± 1.2 |
| 70 to 80 (n = 20) | 74.8 ± 3.2 | 65.2 ± 9.3 | 1.58 ± 0.07 | 25.9 ± 2.7 | 4.6 ± 0.8 |
| | MVV | | | | |
| Men (n = 100) | | | | | |
| 20 to 29 (n = 20) | 21.8 ± 2.3 | 74.5 ± 9.7 | 1.77 ± 0.08 | 23.8 ± 2.4 | 5.0 ± 1.3 |
| 30 to 39 (n = 18) | 34.6 ± 3.4 | 76.5 ± 8.7 | 1.75 ± 0.04 | 25.1 ± 2.6 | 4.8 ± 1.3 |
| 40 to 49 (n = 17) | 44.5 ± 2.6 | 76.1 ± 9 | 1.71 ± 0.07 | 26.1 ± 2.6 | 5.1 ± 1.8 |
| 50 to 59 (n = 15) | 53.6 ± 2.8 | 77.6 ± 11.0 | 1.72 ± 0.06 | 26.1 ± 3.0 | 5.1 ± 1.2 |
| 60 to 69 (n = 16) | 63.9 ± 2.6 | 74.6 ± 9.0 | 1.65 ± 0.09 | 27.4 ± 1.7 | 5.1 ± 1.6 |
| 70 to 80 (n = 14) | 74.6 ± 4.0 | 71.4 ± 7.2 | 1.66 ± 0.08 | 26.0 ± 2.1 | 4.7 ± 1.0 |
| Women (n = 111) | | | | | |
| 20 to 29 (n = 18) | 23.9 ± 3.6 | 59.5 ± 9.6 | 1.65 ± 0.05 | 22.0 ± 2.8 | 4.5 ± 1.0 |
| 30 to 39 (n = 18) | 33.8 ± 3.1 | 62.2 ± 7.3 | 1.63 ± 0.06 | 23.4 ± 2.1 | 4.4 ± 1.2 |
| 40 to 49 (n = 19) | 45.6 ± 3.2 | 65.0 ± 9.8 | 1.61 ± 0.07 | 25.0 ± 3.6 | 4.5 ± 1.4 |
| 50 to 59 (n = 19) | 53.9 ± 3.5 | 63.1 ± 7.0 | 1.59 ± 0.07 | 25.2 ± 2.9 | 4.5 ± 1.3 |
| 60 to 69 (n = 17) | 64.1 ± 3.1 | 64.8 ± 8.5 | 1.57 ± 0.07 | 26.2 ± 2.8 | 4.6 ± 1.2 |
| 70 to 80 (n = 20) | 74.8 ± 3.2 | 65.1 ± 9.3 | 1.58 ± 0.07 | 25.9 ± 2.7 | 4.6 ± 0.8 |

Data presented as mean ± standard deviation. BMI: body mass index. MVV: maximal voluntary ventilation. MRP: maximal respiratory pressure. HPA: Habitual Physical Activity.

**Table 2. Pulmonary function of participants who performed MRP and MVV assessments.**

| Variables | MRP | | MVV | |
|---|---|---|---|---|
| | Men | Women | Men | Women |
| FVC (L) | 4.4 ± 0.8 | 3.1 ± 0.6 | 4.4 ± 0.9 | 3.1 ± 0.6 |
| FVC (% predicted) | 92.1 ± 10 | 95.2 ± 13.6 | 93.1 ± 9.7 | 95.3 ± 11.7 |
| FEV$_1$ (L) | 3.7 ± 0.7 | 2.7 ± 0.6 | 3.7 ± 0.7 | 2.7 ± 0.6 |
| FEV$_1$ (% predicted) | 95.5 ± 11.3 | 99.1 ± 14.3 | 97.3 ± 10.1 | 100.8 ± 13.4 |
| FEV$_1$/FVC ratio | 0.84 ± 0.1 | 0.85 ± 0.1 | 0.84 ± 0.1 | 0.85 ± 0.1 |
| FEV$_1$/FVC ratio (% predicted) | 104 ± 8.3 | 101 ± 12.9 | 104.9 ± 7.7 | 105 ± 8.5 |

Data presented as mean ± standard deviation. FVC: forced vital capacity. FEV$_1$: forced expiratory volume in the first second. MVV: maximal voluntary ventilation. MRP: maximal respiratory pressure. L: liters.

**Table 3. MRP and MVV by age and sex.**

| Age (years) | Men | | | Women | | |
|---|---|---|---|---|---|---|
| | MIP | MEP | MVV | MIP | MEP | MVV |
| | (cmH$_2$O) | (cmH$_2$O) | (L) | (cmH$_2$O) | (cmH$_2$O) | (L) |
| | n = 111 | n = 111 | n = 100 | n = 132 | n = 132 | n = 111 |
| 20 to 29 | 119.2 ± 33.7 | 152.6 ± 47.8 | 168.2 ± 36.0 | 96.9 ± 14 | 117 ± 27.7 | 123.9 ± 14.0 |
| 30 to 39 | 126.8 ± 23.5 | 169.1 ± 38 | 173.3 ± 26.3 | 94.5 ± 18.3 | 112 ± 19.8 | 118.5 ± 25.3 |
| 40to 49 | 115.6 ± 26.3 | 164.1 ± 42.8 | 157.7 ± 25.2 | 90.2 ± 16.7 | 115.1 ± 26.1 | 112.5 ± 16.2 |
| 50 to 59 | 107.6 ± 16.2 | 150.2 ± 26.4 | 158.4 ± 22.0 | 82.7 ± 17 | 95.4 ± 23.7 | 96.6 ± 17.0 |
| 60 to 69 | 91.7 ± 17.7 | 131.8 ± 30.2 | 124.7 ± 23.6 | 83.2 ± 19.7 | 103.3 ± 26.5 | 84.9 ± 15.5 |
| 70 to 80 | 93.9 ± 25.6 | 117.1 ± 18.6 | 109.0 ± 26.3 | 78.1 ± 13.4 | 95.8 ± 17.1 | 85.1 ± 14.7 |
| 20 to 80 | 111.0 ± 28.0 | 149.6 ± 40.3 | 150.6 ± 35.2 | 87.9 ± 17.6 | 106.7 ± 25.2 | 103.4 ± 23.2 |

Data presented as mean ± standard deviation. MIP: maximal inspiratory pressure. MEP: maximal expiratory pressure. MVV: maximal voluntary ventilation. L: liters. cmH$_2$O: centimeters of water.

**Table 4. Prediction equations for MRP and MVV in Brazilian adults.**

| | | Prediction equations | R$^2$ | SEE | LLN |
|---|---|---|---|---|---|
| MIP (cmH$_2$O) | Men | 137−0.57 x (*age*) | 0.13 | 26.11 | MIP$_{predicted}$−42.95 |
| | Women | 107.3−0.4 x (*age*) | 0.16 | 16.10 | MIP$_{predicted}$−26.48 |
| MEP (cmH$_2$O) | Men | 179.9−0.67 x (*age*) | 0.08 | 38.54 | MEP$_{predicted}$−63.4 |
| | Women | 127.4−0.43 | 0.08 | 24.09 | MEP$_{predicted}$−39.63 |
| MVV (L) | Men | 206.3−1.18 x (*age*) | 0.36 | 28.08 | MVV$_{predicted}$−46.19 |
| | Women | 146.3−0.86 x (*age*) | 0.42 | 17.56 | MVV$_{predicted}$−28.89 |

MIP: maximal inspiratory pressure. MEP: maximal expiratory pressure. MVV: maximal voluntary ventilation. L: liters. cmH$_2$O: centimeters of water. R$^2$: coefficients of determination adjusted. SEE: standard error of estimate. LLN: lower limit of normal.

## Discussion

The main findings of this multicentric study was to establish reference values for MIP, MEP, and MVV and propose prediction equations for the healthy Brazilian population aged between 20 and 80 years. The following results were observed: 1) prediction equations were established for MIP, MEP, and MVV in the Brazilian population; 2) age was negatively correlated with MRP and MVV; and 3) men had higher MRP and MVV values than women.

This was the first Brazilian study establishing prediction equations for MRP and MVV based on a multicentric sample, allowing the representation of an ethnically heterogeneous population. Previous studies were conducted in a single Brazilian region [12, 15–17] and may not represent the ethnographic variability from different Brazilian regions.

Brazil is a mixed country, being considered one of the most heterogeneous populations in the world [26]. Being the result of colonization and consequent interethnic crossing between Europeans, represented mainly by the Portuguese, African slaves and native Amerindians [26], in addition to the native population, the indigenous people. This results in variations in skin pigmentation [27]. According to the Brazilian Institute of Geography and Statistics–IBGE [28], in its last census publication, in 2020, Brazil presents a varied racial distribution with 45.35% of Brazilians self-declaring brown, followed by 43.36% white, 10.17%, black, 0.60% indigenous and 0.42% yellow, with the first three classifications covering 99% of the general

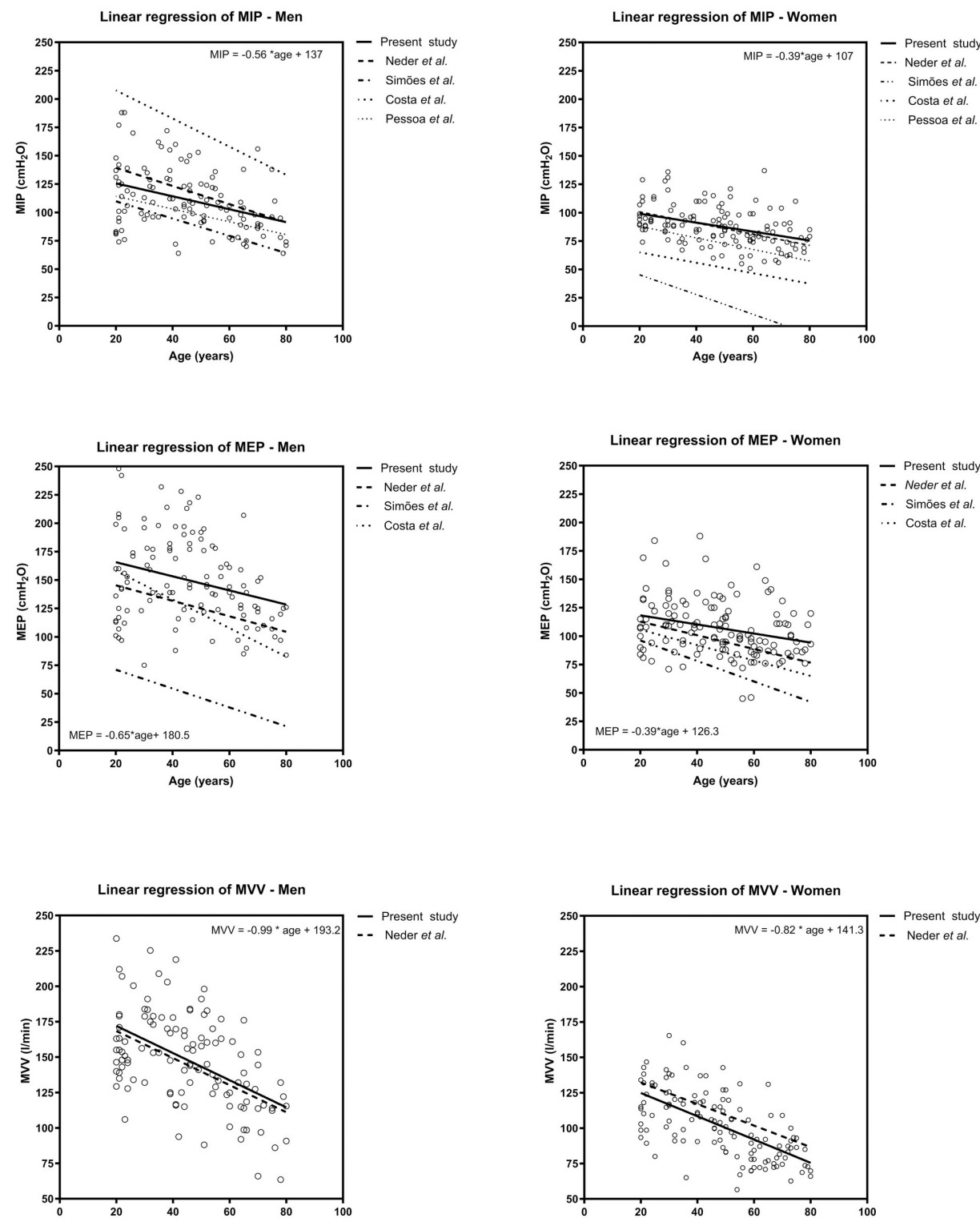

**Fig 2. Line of best fit for values observed in the study and predicted values of maximal respiratory pressure and maximal voluntary ventilation from previous studies.**

population. When analyzing the regions of Brazil, the South has the highest number of self-declared white people with 72.58%, followed by 21.71% mixed race, 5.03% black, 0.40% yellow and 0.24% indigenous; The Southeast is 49.88% white, 38.70% mixed race, 10.61% black, 0.67% yellow and 0.13% indigenous; The Northeast has 59.57% mixed race, 26.66% white, 13.04% black, 0.13% yellow and 0.60% indigenous; The North is 20.74% white, 67.16% brown, 8.82% black, 0.17% yellow and 3.11% indigenous; The Midwest has 37.04% white, 52.40% mixed race, 9.15% black, 0.37% yellow and 1.04% indigenous. Our study evaluated three centers, distributed in 2 regions, Northeast and Southeast, with both having a larger population self-declared as white and mixed race.

The significance of tailoring reference values adapting to the local population cannot be underestimated [29]. When evaluating endurance or strength of respiratory muscles, using standardized values based on data not updated, measured with not recommended devices or from other populations may not accurately reflect the unique characteristics and diversities of the Brazilian population. Elements such as genetics, lifestyle, culture, and environmental influences can exert a substantial effect on the health and performance of individuals [30, 31]. Therefore, it is imperative to establish specific reference values that are tailored to the demographics of a particular region or population [30].

We followed international [5] and national [32] guidelines, which recommend equipment high precision for obtaining MIP, MEP, and MVV measures [18]. Neder et al. [12], Simões et al. [16], and Costa et al. [17] used an aneroid manometer, which has low precision and high variability, different from dedicated software and computers connected to devices based on pressure transducers [5]. Also, Pessoa et al. [15] used a similar methodology to calculate reference values for MIP and MEP in individuals from Minas Gerais state; however, they used waist circumference for prediction equations, hampering their applicability.

Age was negatively correlated with MRP, corroborating other studies [12, 15–17]. MVV was also negatively correlated with age, corroborating studies from Neder et al. [12] and McClaran et al. [33], who observed a 12% reduction of MVV in healthy and active older adults over six years [33]. Aging causes musculoskeletal changes, such as shifts from type II to type I muscle fibers and reduced muscle mass (i.e., sarcopenia) [34], fiber size, strength, and power [35]. In this sense, age can also influence respiratory muscles by reducing diaphragmatic strength due to muscle atrophy and decreased number of type II muscle fibers; therefore, predisposing individuals to diaphragmatic weakness and reduced respiratory muscle strength [36].

Men had higher MRP values than women, corroborating previous studies [12, 15–17]. This result may occur since men have greater muscle mass, number of muscle fibers, and proportion of type II fibers than women; type II fibers have more potential to produce strength and power than type I fibers [35]. Moreover, the zone of apposition is lower, and the diaphragm length is approximately 9% shorter in women than in men. As a result, the diaphragm of women generates low esophageal and gastric pressures during breathing (i.e., low diaphragmatic action) [37]. This may be a complementary mechanism for increased respiratory strength and volume in men.

Most previously published reference values for MRP [12, 15–17] represented the local sample. In this study, both sexes presented higher MIP values than the reference values proposed by Simões et al. [16] and Pessoa et al. [15], (all $p < 0.01$). However, MIP was similar to reference values for women ($p = 0.54$) and lower than values for men ($p = 0.01$) from the study by Neder et al. [12] and higher than the values for men and lower than the values for women in values from Pessoa et al. [15] ($p < 0.01$). MEP in both sexes was higher than the reference values proposed by Costa et al. [17], Simões et al., [16] and Neder et al. [12], (all $p < 0.01$). We could not compare MEP with reference values from Pessoa et al. [15] since they used waist

circumference in the prediction equation. MVV was similar to reference values for men (p = 0.32) and lower than values for women (p = 0.02) from the study by Neder et al. [12].

When comparing the values obtained in our sample to those in other studies [12, 15–17], we found that only the reference values for MIP in women and MVV in men from the study by Neder et al. [12] were consistent with our results. Factors such as measurement methods, different instruments, and individual biological and population characteristics may have influenced the assessment of pulmonary function [18]. For instance, the study by Neder et al. [12] was conducted in 1999, and subsequent studies have not updated prediction equations. The discrepancies between our study's findings and others are primarily attributed to biosocial, sociodemographic, and physical changes (e.g., eating habits, urbanization, and sedentary lifestyles) observed in the Brazilian population. Therefore, it is crucial to update these measurements periodically.

Only Neder et al. [12] developed prediction equations for MVV in the Brazilian adult population; therefore, our comparisons were limited to this study since reference values represent a specific ethnic population. Recently, predictive values for MVV were developed by Silva et al. [38]; however, the authors studied Brazilian individuals aged 6 to 17 years. Prediction equations developed in the present study may represent the Brazilian adult population since they were performed in different locations across the country, reflecting ethnic diversity.

Studies proposing prediction values for pulmonary function must not include individuals with restrictive and obstructive disorders. Among previous studies regarding prediction equations for MRP and MVV in the Brazilian population, only Pessoa et al. [15] adopted pulmonary function values as inclusion criteria. Thus, according to respiratory variables, other studies that suggested reference values may not have included a healthy population, leading to selection bias.

Possessing reference values for MVV that apply to clinical practice is essential. The evaluation of MVV offers supplementary insights and holds clinical relevance for healthy individuals and people with diseases [39]. Furthermore, MVV is employed in several assessments, notably in cardiopulmonary exercise testing, a standard procedure for evaluating exercise capacity. This parameter proves invaluable for quantifying ventilatory reserve in patients dealing with respiratory and cardiovascular conditions [40]. By considering this aspect, healthcare professionals can differentiate between cardiovascular and respiratory profiles in exercise intolerance. However, it is imperative to possess reference values for the ventilatory reserve to enhance the comprehension of the exercise response.

The level of physical activity may have also influenced the results. Only Costa et al. [17] assessed the level of physical activity of the sample for calculating prediction equations. Although this variable was not significantly correlated in the present study, the sample was considered homogeneous since we did not include participants with high physical activity levels. Also, BMI within the normal range was considered an inclusion criterion, whereas Neder et al. [12] included obese participants (i.e., BMI > 29.9 kg/m$^2$). Arenaa and Cahalinb [41] assessed cardiorespiratory fitness and respiratory muscle function in obese subjects (BMI ≥30 kg/m2), linking obesity to respiratory muscle dysfunctions that impact vital capacity. These dysfunctions lead to symptoms of exertion and limitations in functional residual capacity. Other studies have also linked increased weight to a reduction in respiratory muscle strength [42–44], as well as to a decrease in MVV [43]. Our study was designed to establish prediction equations for MRP and MVV, update pre-established values, and correct methodological errors from previous studies.

While our study adhered to a rigorous methodology, it is important to acknowledge several limitations that warrant consideration. Firstly, due to the multicenter nature of the study, evaluators varied among centers, introducing some degree of inter-observer variability.

Additionally, the non-random sampling method employed in this research may limit the generalizability of our findings. Furthermore, it is imperative to highlight the previous period during which data collection occurred, as there was a delay in analyzing the results, which could potentially affect the relevance of the proposed equations. Additionally, it is imperative to highlight the need for additional validation studies encompassing populations from diverse Brazilian regions. Although our study is multicenter in nature, we did not include ethnic variations in our analysis, perhaps limiting the applicability of our prediction equations. Lastly, it is worth noting that our study did not encompass participants aged over 80 years, thereby excluding this segment of the population from our findings. Consequently, the proposed equations may not accurately represent this older demographic. Future research endeavors should strive to bridge this gap by considering the respiratory health of individuals above the age of 80 to provide a more comprehensive insight into this segment of the population.

## Conclusions

In conclusion, the reference values for MIP, MEP, MVV established in this study serve as valuable benchmarks for Brazil's ethnically diverse population. These findings adhere to the guidelines set forth by the European Respiratory Society [5] and the American Thoracic Society / European Respiratory Society (ATS/ERS) [23]. They are not only pertinent in clinical contexts but also serve as essential references for research endeavors involving individuals aged between 20 and 80 years. These standardized values provide a robust foundation for assessing respiratory health and function in a broad cross-section of the Brazilian population, ultimately contributing to more accurate diagnoses and effective treatment strategies.

## Supporting information

**S1 Table. Description of prediction equations for MRP and MVV from previous studies.**
(DOCX)

**S1 Data.**
(XLSX)

## Author Contributions

**Conceptualization:** Palomma Russelly Saldanha Araújo, Jéssica Danielle Medeiros da Fonseca, Marlene Aparecida Moreno, Arméle de Fátima Dornelas de Andrade, Vanessa Regiane Resqueti, Guilherme Augusto de Freitas Fregonezi.

**Data curation:** Palomma Russelly Saldanha Araújo, Ana Aline Marcelino, Guilherme Augusto de Freitas Fregonezi.

**Formal analysis:** Palomma Russelly Saldanha Araújo, Jéssica Danielle Medeiros da Fonseca, Matias Otto Yañez, Rodrigo Torres-Castro, Guilherme Augusto de Freitas Fregonezi.

**Funding acquisition:** Vanessa Regiane Resqueti, Guilherme Augusto de Freitas Fregonezi.

**Investigation:** Marlene Aparecida Moreno, Arméle de Fátima Dornelas de Andrade, Vanessa Regiane Resqueti, Guilherme Augusto de Freitas Fregonezi.

**Methodology:** Palomma Russelly Saldanha Araújo, Guilherme Augusto de Freitas Fregonezi.

**Project administration:** Guilherme Augusto de Freitas Fregonezi.

**Supervision:** Guilherme Augusto de Freitas Fregonezi.

**Writing – original draft:** Palomma Russelly Saldanha Araújo, Guilherme Augusto de Freitas Fregonezi.

**Writing – review & editing:** Jéssica Danielle Medeiros da Fonseca, Ana Aline Marcelino, Marlene Aparecida Moreno, Arméle de Fátima Dornelas de Andrade, Matias Otto Yañez, Rodrigo Torres-Castro, Vanessa Regiane Resqueti, Guilherme Augusto de Freitas Fregonezi.

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
