## [Decision Letter · Decision Letter 0]

6 Jun 2024

PONE-D-24-16298

Reference values for maximal respiratory pressure and maximal voluntary ventilation in the brazilian adult population: a multicentric study

PLOS ONE

Dear Dr.  Fregonezi,

Thank you for submitting your manuscript to PLOS ONE. After careful consideration, we have decided that your manuscript does not meet our criteria for publication and must therefore be rejected.

The main reasons for rejection are as follows: 1.consistency with the reviewer's comments, there are lots important methodological limitations in submission; 2. many similar studies have been reported.

I am sorry that we cannot be more positive on this occasion, but hope that you appreciate the reasons for this decision.

Kind regards,

Yongzhong Guo, Ph.D

Academic Editor

PLOS ONE

Reviewers' comments:

Reviewer's Responses to Questions

**Comments to the Author**

1. Is the manuscript technically sound, and do the data support the conclusions?

Reviewer #1: Yes

Reviewer #2: Yes

Reviewer #3: Yes

2. Has the statistical analysis been performed appropriately and rigorously? 

Reviewer #1: Yes

Reviewer #2: Yes

Reviewer #3: Yes

3. Have the authors made all data underlying the findings in their manuscript fully available?

Reviewer #1: Yes

Reviewer #2: Yes

Reviewer #3: Yes

4. Is the manuscript presented in an intelligible fashion and written in standard English?

Reviewer #1: Yes

Reviewer #2: Yes

Reviewer #3: Yes

5. Review Comments to the Author

Reviewer #1: Dear authors,

Thank you very much for your effort and contribution to area. I have some minor suggestions.

1. On page 3 line 71, you should add methods or devices after recommended.

2. On page 4 line 74 you should add the reference of single center.

3. On page 5 line 116, you should write full name of BMI before abbreviation.

4. On page 7 line 162, you should write full name of HPAI before abbreviation.

5. On page 11 line 222 you should add years after 20-80.

6. I think on page 12 line 231-245, on page 13 line 249-254, should be removed and embedded into discussion.

7. In the discussion on two side, woman should be changed into women. on page 12-13

8. You should say 'the main findings of the our study' rather than the following results in the first paragraph of discussion.

9. I think on page 17 line 361-263 should be removed from limitations. You assessed people from three regions it is not limitation.

10. On page 18 line 378 you should add years after 20-80.

All the best wishes.

Reviewer #2: Thank you very much for inviting me to review this original investigation about new reference equations for maximal respiratory pressures (MRP) and maximal voluntary ventilation (MVV) in Brazilian population. Even when several reference equations for MRP exist worldly and specifically in Brazilian adult population, I do believe this paper add value to the existing literature. Congratulation to the research team for the titanic effort done.

However, some minor and also some major concerns should be answered by the authors.

Introduction

First paragraph (lines 51-54): I think references can be reduced in this paragraph.

Lines 59-60: “While the validity of MVV during a 12 or 15-second test of such brief duration is uncertain in reflecting respiratory endurance”. I think the reference 6 (Laveneziana et al., 2019) should be included here. Indeed, the MVV is not recommended anymore by the ERS as an endurance test. That is why I think authors needs to reinforce the potential value of this new reference equations for MVV, both in the introduction and the discussion.

Lines 66-68: “Although previous studies suggested prediction equations for MIP, MEP, and MVV in the Brazilian population, [6, 9, 12-14]” I think reference number 6 is not adequate here. On the other hand, there are more reference equations for MIP and MEP in Brazilian population (i.e.: Sgariboldi et al. (2015) and Sánchez et al., (2018)). I suggest to include them in the introduction and use then after in the discussion to enrich the discussion. I miss reference equations for MVV in Brazilian population in this paragraph.

Lines 76-78: MIP, MEP and MVV were already expanded above, you don´t need to expand the abbreviations again here.

There is an error concept along the manuscript between “gender” and “sex”. Authors mean sex (as a biological variable) and not gender (which is a social identification). Please, correct this error along the manuscript. According to the lasted statement from ERS for pulmonary function sex and gender should not be confounded in reference equations. https://doi.org/10.1183/13993003.01499-2021

Methods

Firstly, is surprising that data were collected between 2009-11 and not published yet. How authors can explain this fact? If we take into account what ERS says about the obsolescence of reference equations (10 years- Eur Respir J. 2005;26(5):948-68) these data are already obsoleted. How authors can defend these values as representative of the actual population?

Some important details are missing in the methods section: 1) Which was the strategy used for recruitment; where they obtain the sample; the strategy was the same in each research center? 2) Which strategies were used to ensure the consistence of the measurements (did evaluators have some training, some quality control was performed, etc.)?

Line 143: “acceptable when measurements varied less than 5% or were equivalent to 200 mL” [19]. In reference 19 (Graham et al 2019) authors state: “FVC repeatability is achieved when the

difference between the largest and the next largest FVC is <0.150 L for patients older than 6 years of age (86) and <0.100 L or 10% of largest FVC, whichever is greater, for those aged 6 years or younger (8, 87). For FEV1 repeatability, the difference between the largest and the next largest FEV1 is <0.150 L for those older than 6 years of age and <0.100 L or 10% of the largest FEV1, whichever is greater, for those aged 6 years or younger” Please, change the reference or change the numbers.

Line 143-147: how many breaths per minute were aimed for the MVV? In reference number 20 (Miller et al., 2005) 90-110 bpm are remarked as ideal.

Lines 149-158: more technical details are needed. For example: which kind of mouthpiece were used; which pressure was used (peak pressure or plateau pressure- I know that MicroRPM measures the plateau but it should be clarified in the paper); was PUMA software used to see the graphs during the measurements; which quality criteria was used regarding the graph shapes; how much each maneuver lasted?; did the participants rest between maneuvers, and between MIP and MEP?; in which order subjects performed the tests (MIP, MEP and MVV), was always the same?

Lines 160-169: Baecke HPA questionnaire was created for adults over 60 years old, but here it was applied for subjects from 20 years old. Did the authors verify the psychometric proprieties of this questionnaire in the youngest? Could you add some reference?

Line 168 (about algorithm for interpretation) needs a reference.

Results

Line 194: the sample size calculation was 366 subjects but the final sample is composed by 243 for MRP and 211 for MVV. How authors explain so the power calculation of this reference equations if they not achieve the sample size?

Please review table 1: the HPA is under the anthropometric data and this is not correct. Moreover, it is highly important to add the number of subjects (n) achieved in each age group (add this information in the table); only them we can have an idea of the representation in each decade.

I would suggest to merge the pulmonary function data with table 1, after all, these data were collected for characterized the sample, right?

Lines 231-245: authors give p value of their new reference values compared with other reference values published in Brazilian population, but I don’t see this nor as an objective of the study or in the data analysis. Please, add this first as a secondary aim and in the data analysis.

Discussion

I would suggest to enrich the discussion with other Brazilian equations (please see comments in introduction section) and also with the comparation of the most recent MRP equations published. Also, I think it is important to discuss about why only age was included in the equations compared with others that include anthropometric data, as for example the BMI.

Line 345-353: why people with high level of physical activity was not included in the study? I did not see this as a selection criterion. So, is this population really a good representation of Brazilian people? Which are the levels of physical activity in healthy Brazilian population nowadays? Please, justify this.

The LLN was not discussed at all.

The value of reference equations for MVV needs to be reinforce taking into account that this test is not longer recommended in the last statement of ERS (Laveneziana et al., 2019).

Conclusions

Lines 372-372: MIP, MEP and MVV were already expanded above, you don´t need to expand the abbreviation again here.

Reviewer #3: The main aim of this research article is clear. The aim is to obtain reference values for MIP, MEP and MVV in healthy Brazilian subjects. The title should have respiratory muscle strength in place of maximal respiratory pressure.

Although the aim is novel, the study itself is not, as this data has been previously collected in Brazil, but not in multiple sites. The Introduction and Methods section have appropriate detail. It would be good to get a better idea of the ethnicity of the cities and participants, and also previous studies. Results are clear and the data is reasonably well presented with tables and graphs.

The Discussion has a satisfactory level of critical analysis, although more can be added. The significance of the study was given and comparisons made with previous studies, and an emphasis the applied nature of this research.

They are no major issues with the methodology or data analysis. This manuscript is well presented. The language and grammar is acceptable, some improvements in grammar could be made. References are correct and relevant.

Corrections/suggestions for the authors are mentioned below:

Abstract – no changes

Intro

1. Line 55 strength (singular not plural).

2. Line 60. Alter: ‘it is actually the most clinically utilized’ to ‘it is actually clinically utilized the most’.

3. Line 68 – reference 6 is an ERS statement, so how is this a previous study giving Brazilian prediction equations?

4. Line 72-73, can you show where in the ERS statement it mentions a digital manometer is recommended?

Methods

1. Line 95. Give more details on how participants were recruited.

2. Line 96, Brazil has varied ethnicities, so how was this measured and taken into account with the data analysis and regression equations?

3. Line 125 was weight and height taken with or without shoes or coats/jackets?

4. Line 135: …(ATS/ERS) recommendations.

5. Line 143 for MVV assessment, was respiratory rate (RR) measured, as this can affect MVV and were subjects guided to a fixed RR or free breathing RR? See https://www.ncbi.nlm.nih.gov/pmc/articles/PMC5968560/

6. Line 154. Is not a flanged mouthpiece the standard? Could a cylindrical one cause greater leakage?

7. Line 177. so n=366 was the number in total or for each centre?

8. Line 183. Was correlation a linear, logistic or multiple regression?

Results

1. Line 194. 324 were recruited, 244 were included. What happened to the 80 subjects whose data was not collected? Clear from Figure 1, but could say that 80 did not met inclusion criteria.

2. Line 196. Give the reason(s) why 32 could not perform the MVV?

3. Line 199, use term separated and not divided.

4. Line 207/Table 2. Give FVC/FEV1 ratio data to 2 decimal places.

5. Add sub-headings to Results section.

6. Figure 2 – seems images have poor resolution.

7. Did you analyse the differences between the three cities – was there a difference, which might explain the variation in the previous results?

Discussion

1. Line 265 – mention is made of an ethnically heterogenous population, but no evidence in the Methods is given. If this is the case, give an idea of the ethnic background of Brazil and the percentage of your subjects accordingly. Also give the ethnic background of the three different cities.

2. Give more critical analysis on the previous studies – did they report ethnic differences, and what geographical areas did they study, which could have been influenced by ethnicity.

3. Line 279 , where do these recommendations state this.

4. Line 305 – sentence can be written in more clear language.

5. Line 324 – discussion on equipment was given before, so don’t repeat it.

6. Line 351. Does a BMI >25 but <29.9 significantly affect MRP and MVV?

6. PLOS authors have the option to publish the peer review history of their article (what does this mean?). If published, this will include your full peer review and any attached files.

Reviewer #1: **Yes: **Ebru CALIK KUTUKCU

Reviewer #2: No

Reviewer #3: **Yes: **Mirza M F Subhan

- - - - -

---

## [Author Response · Author response to Decision Letter 0]

21 Aug 2024

Yongzhong Guo, Ph.D Academic Editor

Plos One Natal June 25

Subject: Revision and resubmission of Manuscript PONE-D-24-16298

Dear Editor

Thank you for revising our manuscript. We appreciate the reviewers for their complimentary comments and suggestions. We have revised the manuscript in accordance to the recommendations.

Please find attached a point-by-point response to the reviewer’s comments. We hope that you find our answers satisfactory, and that the manuscript is now acceptable for publication.

Sincerely,

Prof. Dr. Guilherme Augusto de Freitas Fregonezi

RESPONSE TO REVIEWERS

Journal Requirements:

REVIEWER 1

Comments to the Author Dear authors,

Thank you very much for your effort and contribution to area. I have some minor suggestions.

1. On page 3 line 71, you should add methods or devices after recommended.

Author’s: Thank you for your note. We've supplemented the information in the introduction section with the following sentence: “Many MIP and MEP studies used less accurate and not recommended mechanical devices, since the nineties, to study respiratory muscle strength” (page 3, line 73 and 74).

2. On page 4 line 74 you should add the reference of single center.

Author’s: We've supplemented the information in the introduction section by adding the following reference: [12] Neder, J.A., et al., Reference values for lung function tests. II. Maximal respiratory pressures and voluntary ventilation. Braz J Med Biol Res, 1999. 32(6):

p. 719-27. (page 4, line 77).

3. On page 5 line 116, you should write full name of BMI before abbreviation.

Author’s: We've supplemented the information in the methods section with the following sentence: “anthropometry data (weight, height, and Body Mass Index – BMI)” (page 5, line 120 and 121).

4. On page 7 line 162, you should write full name of HPAI before abbreviation.

Author’s: We've add the information in the methods section with the following sentence: “The level of physical activity was assessed using the Brazilian version of the Baecke Habitual Physical Activity – HPA questionnaire” (page 8, line 172).

5. On page 11 line 222 you should add years after 20-80.

Author’s: We've supplemented the information in the results section.

6. I think on page 12 line 231-245, on page 13 line 249-254, should be removed and embedded into discussion.

Author’s: Thank you for your note. The supplementary material presented in the results section was added in the discussion section (page 16 and 17, line 325 – 336).

7. In the discussion on two side, woman should be changed into women. on page 12-13.

Author’s: We've corrected English grammatical errors.

8. You should say 'the main findings of the our study' rather than the following results in the first paragraph of discussion.

Author’s: We've corrected the sentence in the discussion section.

9. I think on page 17 line 361-263 should be removed from limitations. You assessed people from three regions it is not limitation.

Author’s: We have excluded the sentence from the discussion section.

10. On page 18 line 378 you should add years after 20-80.

Author’s: Thank you the note. We've supplemented the information in the conclusions section.

REVIEWER 2

Comments to the Author

Thank you very much for inviting me to review this original investigation about new reference equations for maximal respiratory pressures (MRP) and maximal voluntary ventilation (MVV) in Brazilian population. Even when several reference equations for MRP exist worldly and specifically in Brazilian adult population, I do believe this paper add value to the existing literature. Congratulation to the research team for the titanic effort done. However, some minor and also some major concerns should be answered by the authors.

Author’s: We appreciate your contributions and have carefully reviewed our manuscript, addressing each of your points. We believe it is now ready for publication.

Introduction

1. First paragraph (lines 51-54): I think references can be reduced in this paragraph.

Author’s: Thank you for your note. We removed the reference Rocha et al. (2007), by repeating information.

2. Lines 59-60: “While the validity of MVV during a 12 or 15-second test of such brief duration is uncertain in reflecting respiratory endurance”. I think the reference 6 (Laveneziana et al., 2019) should be included here. Indeed, the MVV is not recommended anymore by the ERS as an endurance test. That is why I think authors needs to reinforce the potential value of this new reference equations for MVV, both in the introduction and the discussion.

Author’s: Thank you for your comment. We've supplemented the reference, and we additionally complemented the information in the introduction section:

“Respiratory muscle endurance is assessed using maximal voluntary ventilation (MVV).[5] The validity of MVV in assessing respiratory endurance during brief 12 or 15-second test is uncertain.[6] Nevertheless, this maneuver assesses maximum ventilatory capacity, reflecting the functioning of the inspiratory pump and chest wall.[7] It is widely utilized clinically to determine ventilatory reserve,[8, 9] assess risk of postoperative complications[10] and establish targets for muscle training.[11]” (page 3, line 57-63).

7. Colwell KL, Bhatia R. Calculated versus Measured MVV-Surrogate Marker of Ventilatory Capacity in Pediatric CPET. Med Sci Sports Exerc. 2017 Oct;49(10):1987-1992. doi: 10.1249/ MSS.0000000000001318. Erratum in: Med Sci Sports Exerc. 2018 Feb;50(2):390. doi: 10.1249/ MSS.0000000000001521. PMID: 28489684.

8. American Thoracic Society; American College of Chest Physicians. ATS/ACCP Statement on cardiopulmonary exercise testing. Am J Respir Crit Care Med. 2003 Jan 15;167(2):211-77. doi: 10.1164/rccm.167.2.211. Erratum in: Am J Respir Crit Care Med. 2003 May 15;1451-2. PMID: 12524257.

9. Arena R, Sietsema KE. Cardiopulmonary exercise testing in the clinical evaluation of patients with heart and lung disease. Circulation. 2011 Feb 15;123(6):668-80. doi: 10.1161/CIRCULATION AHA.109.914788. PMID: 21321183.

10. Bevacqua BK. Pre-operative pulmonary evaluation in the patient with suspected respiratory disease. Indian J Anaesth. 2015 Sep;59(9):542-9. doi: 10.4103/0019-5049.165854. PMID: 26556912; PMCID: PMC4613400.

11. Markov G, Spengler CM, Knöpfli-Lenzin C, Stuessi C, Boutellier U. Respiratory muscle training increases cycling endurance without affecting cardiovascular responses to exercise. Eur J Appl Physiol. 2001 Aug;85(3-4):233-9. doi: 10.1007/s004210100450. PMID: 11560075.

3. Lines 66-68: “Although previous studies suggested prediction equations for MIP, MEP, and MVV in the Brazilian population, [6, 9, 12-14]” I think reference number 6 is not adequate here. On the other hand, there are more reference equations for MIP and MEP in Brazilian population (i.e.: Sgariboldi et al. (2015) and Sánchez et al., (2018)). I suggest to include them in the introduction and use then after in the discussion to enrich the discussion. I miss reference equations for MVV in Brazilian population in this paragraph.

Author’s: Thank you for the note. We have excluded the reference in the introduction section. We do not included reference from Sánchez et al., (2018) and Sgariboldi et al. (2015), because this article focus on reference values for obese subjects. Regarding study of Sgariboldi et al. (2015), 76% of the subjects were include on overweight, obese or morbidity obesity.

4. Lines 76-78: MIP, MEP and MVV were already expanded above, you don´t need to expand the abbreviations again here.

Author’s: We've corrected the sentence in the introduction section.

5. There is an error concept along the manuscript between “gender” and “sex”. Authors mean sex (as a biological variable) and not gender (which is a social identification). Please, correct this error along the manuscript. According to the lasted statement from ERS for pulmonary function sex and gender should not be confounded in reference equations. https://doi.org/ 10.1183/13993003.01499-2021.

Author’s: We've corrected the sentence along the manuscript.

Methods

1. Firstly, is surprising that data were collected between 2009-11 and not published yet. How authors can explain this fact? If we take into account what ERS says about the obsolescence of reference equations (10 years- Eur Respir

J. 2005;26(5):948-68) these data are already obsoleted. How authors can defend these values as representative of the actual population?

Author: Thank you for your comment. In fact, the period during which data acquisition occurred was extensive. We experienced a delay in obtaining the analysis of the results because it was carried out across multiple centers and involved several collaborators. This is a limitation of the study, a point that will be emphasized in the discussion section. However, we believe in the potential for publishing our results, primarily due to the scarcity of studies in this field during the current period and the rigorous methodology with which the study was conducted. We additionally complemented the information in the conclusion section:

“Furthermore, it is imperative to highlight the previous period during which data collection occurred, as there was a delay in analyzing the results, which could potentially affect the relevance of the proposed equations.” (page 19, line 394-396).

2. Which was the strategy used for recruitment; where they obtain the sample; the strategy was the same in each research center?

Author’s: We have added a methods section - please see below:

“Participants were recruited by convenience through publicity to university students at each center, as well as via social media.” (page 5, line 98 and 99)

3. Which strategies were used to ensure the consistence of the measurements (did evaluators have some training, some quality control was performed, etc.)?

Author: All evaluators underwent prior training to ensure the reproducibility of the study and familiarize themselves with the materials and methods. Finally, the databases were populated, and a joint analysis of the results was conducted. For clarity, we have included the following information in the Methods section: "The same previously trained evaluator at each research center conducted both stages on the same day." (page 6, line 125 and 126).

4. Line 143: “acceptable when measurements varied less than 5% or were equivalent to 200 mL” [19]. In reference 19 (Graham et al 2019) authors state: “FVC repeatability is achieved when the difference between the largest and the next largest FVC is <0.150 L for patients older than 6 years of age (86) and

<0.100 L or 10% of largest FVC, whichever is greater, for those aged 6 years or younger (8, 87). For FEV1 repeatability, the difference between the largest and the next largest FEV1 is <0.150 L for those older than 6 years of age and <0.100

L or 10% of the largest FEV1, whichever is greater, for those aged 6 years or younger” Please, change the reference or change the numbers.

Author’s: Thank you for your considerations. We've corrected the sentence in the results section: "The assessment was repeated three to five times after one-minute intervals and considered acceptable when the difference between the two largest FVC and FEV1 values varied less than ≤0.150 L." (page 7, line 148 and 149).

5. Line 143-147: how many breaths per minute were aimed for the MVV? In reference number 20 (Miller et al., 2005) 90-110 bpm are remarked as ideal.

Author’s: The equipment used follows the technically accepted maneuver according to the ERS/ATS guidelines, the best, traces, were selected automatically based on this criterion. The breathing frequency is approximately 90 breaths per minute, following the reference of Miller MR, Pincock AC. Linearity and temperature control of the Fleisch pneumotachograph. J Appl Physiol 1986;60:710–715.

6. Lines 149-158: more technical details are needed. For example: which kind of mouthpiece were used; which pressure was used (peak pressure or plateau pressure- I know that MicroRPM measures the plateau but it should be clarified in the paper); was PUMA software used to see the graphs during the measurements; which quality criteria was used regarding the graph shapes; how much each maneuver lasted?; did the participants rest between maneuvers, and between MIP and MEP?; in which order subjects performed the tests (MIP, MEP and MVV), was always the same?

Author’s: Thank you for the note. We have added a methods section - please see below:

Study design “The order of respiratory assessments was randomized, respecting the 20- minute interval between them.” (page 6, line 123 - 125)

Respiratory muscle strength “The PUMA PC software (Micro Medical, Rochester Kent, UK) operationalized the MIP and MEP (equivalent to respiratory muscle strength) variables using the aforementioned maximum mean pressures. Participants performed a maximum inspiration from residual volume for MIP and a maximum expiration from total lung capacity for MEP, performing 3 - 5 maneuvers, with a 1 minute break between them. Both were assessed with participants using a disposable flanged mouthpiece and a nose clip.” (page 7, line 159 - 165)

7. Lines 160-169: Baecke HPA questionnaire was created for adults over 60 years old, but here it was applied for subjects from 20 years old. Did the

authors verify the psychometric proprieties of this questionnaire in the youngest? Could you add some reference?

Author’s: Thank you for your considerations. The reference cited in the methodology, Florindo and Latorre (2003), provides validation and reproducibility of Baecke's questionnaire for the Brazilian adult population. In their study, these authors evaluated a population with an average age of 32.6 ± 3.2 years. To assess the validation of the Baecke HPA questionnaire, they measured maximum oxygen consumption (VO2 max), percent decrease in heart rate (%DHR) using the Cooper 12-minute walk or run test, annual physical exercise index (IPE), and weekly locomotion activity index (ILA). They verified reliability through test-retest with a 45- day interval. The authors concluded that Baecke's HPA questionnaire is valid and reliable for measuring habitual physical.

8. Line 168 (about algorithm for interpretation) needs a reference. Author’s: Thank you for your comment. We've supplemented the reference. Results

1. Line 194: the sample size calculation was 366 subjects but the final sample is composed by 243 for MRP and 211 for MVV. How authors explain so the power calculation of this reference equations if they not achieve the sample size?

Author’s: The initial sample size was 183 participants. To mitigate potential data acquisition errors, we increased the sample size to a final total of 366 participants. We evaluated 243 subjects, which falls within the predefined range for our study.

2. Please review table 1: the HPA is under the anthropometric data and this is not correct. Moreover, it is highly important to add the number of subjects (n) achieved in each age group (add this information in the table); only them we can have an idea of the representation in each decade.

Author’s: We have edited the table 1 in the results section and added the sample size (n) to each age group. Regarding the HPA, we have retained it, as it was used as an inclusion criterion and for sample characterization.

3. I would suggest to merge the pulmonary function data with table 1, after all, these data were collected for characterized the sample, right?

Author’s: While your point about it being a characterization variable is valid, we opted to present the anthropometric and pulmonary function data in separate tables. This decision stems from the fact that our spirometric analysis predictive equation is already age-adjusted, making decade-by-decade analysis unnecessary

4. Lines 231-245: authors give p value of their new reference values compared with other reference values published in Brazilian population, but I don’t see this nor as an objective of the study or in the data analysis. Please, add this first as a secondary aim and in the data analysis.

Author’s: We've supplemented the information in introduction section - please see below: “In addition, we will compare the values resulting from our generated equation with those

---

## [Decision Letter · Decision Letter 1]

22 Oct 2024

REFERENCE VALUES FOR RESPIRATORY MUSCLE STRENGTH AND MAXIMAL VOLUNTARY VENTILATION IN THE BRAZILIAN ADULT POPULATION: A MULTICENTRIC STUDY

PONE-D-24-16298R1

Dear Dr. Fregonezi,

We’re pleased to inform you that your manuscript has been judged scientifically suitable for publication and will be formally accepted for publication once it meets all outstanding technical requirements.

Kind regards,

Ming-Ching Lee

Academic Editor

PLOS ONE

Additional Editor Comments (optional):

Reviewers' comments:

Reviewer's Responses to Questions

**Comments to the Author**

1. If the authors have adequately addressed your comments raised in a previous round of review and you feel that this manuscript is now acceptable for publication, you may indicate that here to bypass the “Comments to the Author” section, enter your conflict of interest statement in the “Confidential to Editor” section, and submit your "Accept" recommendation.

Reviewer #1: All comments have been addressed

Reviewer #2: All comments have been addressed

2. Is the manuscript technically sound, and do the data support the conclusions?

Reviewer #1: Yes

Reviewer #2: Yes

3. Has the statistical analysis been performed appropriately and rigorously? 

Reviewer #1: Yes

Reviewer #2: Yes

4. Have the authors made all data underlying the findings in their manuscript fully available?

Reviewer #1: Yes

Reviewer #2: Yes

5. Is the manuscript presented in an intelligible fashion and written in standard English?

Reviewer #1: Yes

Reviewer #2: Yes

6. Review Comments to the Author

Reviewer #1: Dear authors

thank you very much for your effort. All my concerns were adressed now. The article will be useful for interpretation of data in brazilian population

Sincerely

Reviewer #2: Based on my review, I find the revisions made by the authors satisfactory and recommend the paper for acceptance

7. PLOS authors have the option to publish the peer review history of their article (what does this mean?). If published, this will include your full peer review and any attached files.

Reviewer #1: No

Reviewer #2: No

---

## [Editor Report · Acceptance letter]

11 Nov 2024

PONE-D-24-16298R1 

PLOS ONE

Dear Dr. Fregonezi, 

I'm pleased to inform you that your manuscript has been deemed suitable for publication in PLOS ONE. Congratulations! Your manuscript is now being handed over to our production team.

Kind regards, 

on behalf of

Dr. Ming-Ching Lee 

Academic Editor

PLOS ONE